# *Nkx2.9* Contributes to Mid-Hindbrain Patterning by Regulation of mdDA Neuronal Cell-Fate and Repression of a Hindbrain-Specific Cell-Fate

**DOI:** 10.3390/ijms222312663

**Published:** 2021-11-23

**Authors:** Willemieke M. Kouwenhoven, Lars von Oerthel, Maria Gruppilo, Jianmin Tian, Cindy M. R. J. Wagemans, Imke G. J. Houwers, Joseph Locker, Simone Mesman, Marten P. Smidt

**Affiliations:** 1Swammerdam Institute for Life Sciences, University of Amsterdam, 1098 XH Amsterdam, The Netherlands; willemieke.kouwenhoven@umontreal.ca (W.M.K.); larsvonoerthel@hotmail.com (L.v.O.); m.r.j.wagemans@uva.nl (C.M.R.J.W.); imkehouwers@hotmail.com (I.G.J.H.); s.mesman@uva.nl (S.M.); 2Department of Pathology, University of Pittsburgh, Pittsburgh, PA 15260, USA; MGruppilo@pitt.edu (M.G.); Jtian@pitt.edu (J.T.); jlocker@pitt.edu (J.L.); 3Faculty of Pharmacy, University of Montreal, Montreal, QC H3T 1J4, Canada

**Keywords:** *Nkx2.9*, *Nkx2.8*, mdDA, IsO, mouse, *Wnt8b*, mid-hindbrain

## Abstract

*Nkx2.9* is a member of the NK homeobox family and resembles *Nkx2.2* both in homology and expression pattern. However, while *Nkx2.2* is required for development of serotonergic neurons, the role of *Nkx2.9* in the mid-hindbrain region is still ill-defined. We have previously shown that *Nkx2.9* expression is downregulated upon loss of *En1* during development. Here, we determined whether mdDA neurons require *Nkx2.9* during their development. We show that *Nkx2.9* is strongly expressed in the IsO and in the VZ and SVZ of the embryonic midbrain, and the majority of mdDA neurons expressed *Nkx2.9* during their development. Although the expression of *Dat* and *Cck* are slightly affected during development, the overall development and cytoarchitecture of TH-expressing neurons is not affected in the adult *Nkx2.9*-depleted midbrain. Transcriptome analysis at E14.5 indicated that genes involved in mid- and hindbrain development are affected by *Nkx2.9*-ablation, such as *Wnt8b* and *Tph2*. Although the expression of *Tph2* extends more rostral into the isthmic area in the *Nkx2.9* mutants, the establishment of the IsO is not affected. Taken together, these data point to a minor role for *Nkx2.9* in mid-hindbrain patterning by repressing a hindbrain-specific cell-fate in the IsO and by subtle regulation of mdDA neuronal subset specification.

## 1. Introduction

NK homeobox genes, first described in drosophila, program organogenesis during embryonic development [1]. The murine NK2 family consists of seven genes: *Nkx2.1*–*Nkx2.6* and *Nkx2.9*, which is named *Nkx2.8* in most other species [2,3,4]. These seven genes can be subdivided based on their homology to the drosophila NK genes and their dependence on sonic hedgehog (SHH) signaling. *Nkx2.1*, *Nkx2.2* and *Nkx2.9* resemble NK2 drosophila genes and are dependent on SHH-signaling for their expression, whilst *Nkx2.3*, *Nkx2.5* and *Nkx2.6* resemble NK3 and are SHH-independent [4,5]. 

Solely, *Nkx2.1*, *Nkx2.2* and *Nkx2.9* are expressed in the central nervous system. While *Nkx2.1* is involved in ventral programming of the basal ganglia [6], *Nkx2.2* is pivotal for the development of serotonergic neurons in the hindbrain [7]. *Nkx2.9* displays strong similarity both in homology (~88%) and in expression pattern to *Nkx2.2*. During early embryonic development (after E9.5) *Nkx2.9* is present in the dorsal nervous system and is present in the zona limitans intrathalamica (ZLI), which is marked by expression of *Shh* [8]. Nonetheless, *Nkx2.9* is unable to rescue the serotonin neurons in *Nkx2.2*-ablated animals, suggesting it has a unique (though unknown) role during the development of the central nervous system [4,7,9,10,11]. 

Mesodiencephalic dopaminergic neurons (mdDA) are generated around E11.5 and develop into the substantia nigra pars compacta (SNc) and ventral tegmental area (VTA) [12]. The dopaminergic progenitor pool is dependent on the expression of early transcription factors (e.g., *Nurr1*, *Lmx1a/b*, *En1*, *Otx2*), morphogenetic factors *Shh* and Wnt/β-catenin, as well as the correct organization of molecular signaling centers such as the isthmic organizer (IsO) [13,14,15]. Previously, we investigated the role of homeobox transcription factor Engrailed-1 (*En1*) in the development of mdDA neurons, and genome-wide expression analysis on *En1*-ablated dissected midbrain material at E13.5 designated *Nkx2.9* as a direct or indirect transcriptional target of *En1* [16]. Here, we elaborate on this finding and determine whether mdDA neurons require *Nkx2.9* during their developmental progression. We show for the first time that *Nkx2.9* is expressed in the ventricular zone (VZ) and subventricular zone (SVZ) of the midbrain, harboring mdDA progenitor cells, and that the majority of mdDA neurons express *Nkx2.9* during their development. Interestingly, the neuroanatomical architecture and number of TH-neurons is not clearly affected during development, and later, in the adult brain. However, expression of *Dat* and *Cck* appears to be affected in the *Nkx2.9* mutant at E14.5, which recovers in the adult, suggesting a small role for *Nkx2.9* in terminal differentiation. Transcriptome analysis of E14.5 dissected midbrain material of the *Nkx2.9* mutant indicated that early factors in mid- and hindbrain development are alternatively expressed, such as *Wnt8b* in the midbrain and *Tph2* in the hindbrain. Although the expression of *Tph2* extends more rostral into the isthmic area in the *Nkx2.9* mutants, the establishment of the IsO is not affected. Taken together, *Nkx2.9* is an early transcription factor that is expressed by the majority of mdDA neurons during development and has a minor role in mid-hindbrain patterning by repressing a hindbrain-specific cell-fate in the IsO, and by subtle regulation of mdDA neuronal subset specification.

## 2. Results

### 2.1. Nkx2.9 Transcript Expression in Mesencephalon and Metencephalon Is Dependent on En1

The embryonic development of mdDA neuronal subsets depends on the interplay of different (homeobox) transcription factors such as *Pitx3*, *Nurr1*, *Otx2* and *En1* [13,15]. The latter has been shown to be pivotal for the correct development and survival for mdDA neurons [16,17,18] as well as the correct maintenance of the IsO [19]. In a previously published genome-wide expression analysis on E13.5 *En1*-ablated dissected midbrain material, *Nkx2.9* was significantly down-regulated [16], suggesting that *Nkx2.9* is a direct or indirect transcriptional target of *En1*. To substantiate this finding, we performed in situ hybridization of *Nkx2.9* in control and *En1*KO animals (Figure 1). At E12.5 the *Nkx2.9* transcript is present in the rostral hindbrain, the dorsal midbrain, and extends into the diencephalon through expression in the ZLI (Figure 1B). Furthermore, *Nkx2.9* is strongly expressed in the region of the IsO. In the absence of *En1* the expression in medial sections is lost in the IsO-region (asterisk, Figure 1B), but expression is retained in the ZLI. These data indicate that the correct expression pattern of *Nkx2.9* is dependent on *En1* activity, and these data also propose the possibility that *Nkx2.9*-deficiency might contribute to the changes observed in the mid- and hindbrain region as a consequence of *En1*-ablation [19].

### 2.2. Nkx2.9 Transcript Expression Is Present in the mdDA Precursor Region

In order to investigate if *Nkx2.9* has a potential role in the generation and development of mdDA neurons, we elected to first describe the expression pattern at E12.5 relative to the expression pattern of tyrosine hydroxylase (TH), the rate-limiting enzyme in the production of dopamine (Figure 2A,B). Double-labeling of the *Nkx2.9* transcript (blue) and TH protein (brown) at E12.5 revealed that *Nkx2.9* is expressed in the VZ and SVZ of the midbrain. Furthermore, in (para-)medial sections the *Nkx2.9* transcript is expressed in such proximity to the TH protein that possible colocalization cannot be excluded. In contrast, the *Nkx2.9* transcript is clearly located dorsal-caudal to the TH expression pattern in lateral sections. Moreover, in medial sections of the E12.5 developing brain *Nkx2.9* expression is present in the IsO, as is indicated by overlapping expression with *Fgf8* (arrowhead, Figure 2C).

To determine whether *Nkx2.9* could also have a function during terminal differentiation of the mdDA neurons, we visualized *Nkx2.9* mRNA at E14.5. This revealed that the area of *Nkx2.9* expression is strongly reduced compared to its expression area at E12.5. At E14.5 the expression of *Nkx2.9* is restricted to a small region in the mid-hindbrain area at this stage, as *Nkx2.9* mRNA is solely present at the most caudal border of the midbrain (arrowhead, Figure 2D). Thus, based on its temporal and spatial expression pattern *Nkx2.9* appears to be only scarcely expressed in the midbrain area at the time-point at which mdDA neurons terminally differentiate. The abundant expression of *Nkx2.9* in the presumed mdDA precursor zone at E12.5 [20,21], and strong reduction in expression at later time-points, suggests that *Nkx2.9* is most likely able to exert an effect on the dopamine system during the precursor state of mdDA neuronal development. 

### 2.3. A Major Subset of mdDA Neurons Originates from a Nkx2.9-Positive Lineage

To further determine the possible role of *Nkx2.9* in mdDA precursors we analyzed an *Nkx2.9* knock-out/LacZ knock-in a mouse model [22]. In this model, the first exon of the *Nkx2.9* transcript was replaced by a LacZ-NLS which was joined to the ATG start codon at the intact 5-UTR, and *LacZ* expression generally recapitulates endogenous *Nkx2.9* expression as shown previously [22]. As such, we used this model to fate-map cells that at one point in time would have expressed *Nkx2.9* during normal murine development.

First, at E12.5, β-galactosidase (β-gal) protein is present in the IsO (arrowhead, Figure 3B), in the ventricular zone (arrow, Figure 3B) and in the hindbrain (>, Figure 3B). This pattern at this (early) stage largely overlaps with the native *Nkx2.9* expression (compare Figure 2B and Figure 3B). Second, at E14.5 the mid- and hindbrain area harbors cells over the entire lateral-medial axis that are positive for β-gal protein, particularly in the region of the mdDA neuronal pool (arrows, Figure 3D). This is even more apparent in more lateral sections, in which the β-gal protein can be found in the ventral midbrain abutting the mesencephalic flexure (arrows, Figure 3D). Thus, these fate-mapping experiments reveal that during development many cells located in the mid-hindbrain region encountered (transient) expression of *Nkx2.9*. Interestingly, the observed expression of the β-gal protein at E14.5 is much broader than the previously described expression of the *Nkx2.9* transcript, indicating that the expression of the β-gal protein is highly stable and detectable long after expression of *Nkx2.9* has stopped.

Since we observed scarce *Nkx2.9* transcript expression in the region of the mdDA neuronal pool at E14.5 (Figure 2D), but an abundance of β-gal protein in the ventricular zone at E12.5, as well as in the region of the mdDA neuronal pool at E14.5, we hypothesized that a subset of mdDA neurons may have expressed *Nkx2.9* during their development. Thus, we performed a double-labeling experiment to visualize both TH (green) protein and β-gal protein (red) at the stage of terminal differentiation (E14.5) of mdDA neurons. Double-positive mdDA neurons were found both in the rostral-lateral midbrain (Figure 3F, yellow arrowheads, box 1, 3) and in the caudomedial midbrain (Figure 3F, box 5–8). Indeed, this suggests that most of the mdDA neurons (transiently) express *Nkx2.9* at one point in time, considering the expression of *Nkx2.9* transcript these neurons most likely expressed *Nkx2.9* during the precursor stage. Remarkably, the mdDA neurons that are negative for β-gal (Figure 3F, white arrowheads, box 1–8) are interspersed with double-positive mdDA neurons (Figure 3F, yellow arrowheads, box 1–8), indicating that the mdDA neuronal population contains at least two different neuronal populations: a population that expressed *Nkx2.9* during development and a population that did not express this gene. In order to substantiate this point, we analyzed adult *Nkx2.9*LacZ/+ brains for TH and β-gal presence (Figure 3H). This analysis confirmed that double-positive mdDA neurons were found in both the SNc and VTA region neurons (yellow arrowheads, Figure 3H), again in close proximity to mdDA neurons that did not originate from an *Nkx2.9*-positive lineage (white arrowheads, Figure 3H). In sum, we propose that a large subset of mdDA neurons in both the VTA and SNc originates from a Nkx2.9-positive precursor lineage population.

### 2.4. Nkx2.9-Ablation Does Not Affect the mdDA Precursor Proliferation, but Decreases Dat Expression at E14.5

Having demonstrated that a large part of the mdDA neurons finds its origin in an *Nkx2.9*-positive precursor lineage, we aimed to determine the specific role of *Nkx2.9* in these neurons. To start, we analyzed whether the loss of *Nkx2.9* affected the cytoarchitecture of the mid-hindbrain region, and thus we traced LacZ-expressing cells by visualizing β-gal activity in *Nkx2.9*LacZ/+ and *Nkx2.9*LacZ/LacZ hemispheres at E14.5 (Figure 4A–D). In the absence of *Nkx2.9*, the mid- and hindbrain area displayed remarkably increased levels of β-gal activity, compared to the loss of just one allele (arrowheads, Figure 4B, and compare Figure 3F to Figure 4D). This could be indicative of increased proliferation in the isthmic area in the absence of *Nkx2.9*, which may affect the initial pool size of neurons in the mid-hindbrain region. Thus, in order to determine whether proliferation is affected in the *Nkx2.9*LacZ/LacZ embryonic midbrain, we performed an immunohistochemistry experiment for phosphorylated Histone-H3 (pHH3), a marker for mitotic cells [23]. We observed no clear difference in proliferation in the absence of *Nkx2.9* at the stages E12.5 (Figure 4E,F) and E14.5 (Figure 4G,H). The increase in β-gal protein in the mutant would, therefore, most likely be caused by the presence of two functional alleles of LacZ, which would double the levels of β-gal protein in each *Nkx2.9*-fate-mapped cell, rather than by a change in proliferation.

Interestingly, the β-gal level within the mdDA region of *Nkx2.9*-ablated mice displayed diminished expression (arrows, compare Figure 3F to Figure 4D), relative to the levels of β-gal protein expression in the ventricular zone (arrowheads, Figure 4D). This change might suggest that the ablation of *Nkx2.9* affects the developing mdDA neurons without altering proliferation in the mid-hindbrain region. To determine whether the decrease in *LacZ* expression is reflected in a cell-intrinsic decrease or regional-specific loss of mdDA neuronal markers, we performed an independent set of ISH experiments (Figure 5A–C). This showed that expression of *Dat* was affected in absence of *Nkx2.9* and revealed that the loss of *Dat* was highly specific to the most caudomedial region of the midbrain (arrowhead, Figure 5B). In situ hybridization experiments on an adjacent set for *Th* transcript, clarified that the absence of *Dat* was not caused by the selective absence of mdDA neurons in that region (Figure 5C). Since we previously determined that the expression of *Dat* is enriched in a rostral subset of the mdDA neuronal pool [24], we expected that more subset markers might be affected as a consequence of *Nkx2.9* removal. Thus, we investigated the expression patterns of two known markers that define the rostral and caudal subsets *Ahd2* and *Cck*, respectively [15,16]. We observed no clear alterations in the transcript pattern of *Ahd2* on adjacent sections in wild type and *Nkx2.9*-ablated animals (Figure 5D), though *Cck* expression appeared to be slightly affected (Figure 5E, arrowheads). Taken together, these data point to a possible role for *Nkx2.9* in fate determination of mdDA neurons early during development. 

### 2.5. Loss of Nkx2.9 Does Not Affect the mdDA Neuronal Population in the Adult Midbrain

As was shown previously, the *En1* mutant shows a progressive loss of mdDA neurons when animals age [17]. *Nkx2.9* is lost in the *En1* mutant (Figure 1) and the affected fate determination at E14.5 (Figure 5) could similarly result in a loss of mdDA neurons in the adult midbrain. Thus, to determine the consequence of *Nkx2.9* ablation on the size of the mdDA neuronal pool we examined the adult (2.5 months old) midbrain and striatum of *Nkx2.9*-ablated mice and age-matched wild type controls by means of immunohistochemistry for TH (Figure 6). We observed no change in the cytoarchitecture or number of the mdDA neurons at this stage, both in the VTA and in the SNc (Figure 5A,B), which was confirmed by quantification of the TH-expressing cells (Figure 6C, *n* = 3 both WT and Mt). Moreover, no overt changes were detected in the innervation of the striatum (Figure 6D,E). Similar results were found for the presence of *Dat* and *Th* transcripts in the adult midbrain (Appendix A), although *Dat* expression was found to be affected at E14.5. In sum, the loss of *Dat* expression at embryonic stages was restored in the adult and the ablation of *Nkx2.9* did not significantly affect the number of mdDA neurons present in the adult midbrain. 

### 2.6. Loss of Nkx2.9 Results in a Minor Loss of Specification Factors in the Mid- and Hindbrain, Suggesting a Role in Mid- and Hindbrain Fate Determination

In order to analyze the molecular changes upon *Nkx2.9*-ablation in terms of developmental programming of the midbrain area, we performed next-generation RNA sequencing (RNA seq) on micro-dissected midbrains at E14.5 of *Nkx2.9*+/+ and *Nkx2.9*LacZ/LacZ embryos (Figure 7A). Two dissected midbrains per genotype were pooled to create one indexed library, and in total six libraries were produced (*n* = 3 per genotype), which yielded after sequencing between 2–8 million reads per library. The RNA-sequencing data were analyzed with a Dseq2 analysis for determining differential expressed genes. From this analysis we found 287 differential regulated genes with a (wald stats) *p*-value of <0.05. No genes were significantly regulated when using adjusted *p*-values.

We decided to continue with genes that showed a significant change with *p* < 0.01 before post hoc analysis (Appendix A). A panther overrepresentation test to determine whether specific biological processes were enriched in this dataset did not show an overrepresentation of any specific processes. However, from this dataset we did find five interesting target genes based on function during mid- and hindbrain development and expression pattern as shown in the Allen Brain Atlas, namely *En1*, *En2*, *Pax7*, *Tph2*, and *Wnt8b*. Expression of all genes was found to be upregulated, besides *Wnt8b,* which showed a downregulation in the RNA-sequencing analysis (Figure 7A). *En1*, *En2*, and *Wnt8b* have previously been linked to the development of mdDA neurons and cytoarchitecture of the midbrain [17,19,25]. Besides their function in development of the DA system, *En1* and *Wnt8b* have also been implicated in the establishment of the IsO and development of the serotonin system [19,26,27], whereas *Pax7*, and *Tph2* have both been implicated in the development of the noradrenergic and serotonin system of the hindbrain [28,29,30]. The differential regulation of genes involved in the development of mid- and hindbrain-specific neuronal systems and genes important for the establishment of the IsO, points to a possible function of *Nkx2.9* in development of these brain areas, similar to the previously described function of *Nkx2.2* [7,31]. In order to determine whether the differential expression of these genes is due to cell intrinsic changes in expression or a change in spatial expression, we performed in situ hybridization studies on *Nkx2.9* mutants at E14.5. 

In situ hybridization of *Nkx2.9*+/+ and the *Nkx2.9*LacZ/LacZ at E14.5 showed no overt changes in the expression of *En1* (Figure 7B), *Pax7* (Figure 7C), and *En2* (Figure 7D), indicating that the changes found in the transcriptome analysis were not reflected in an altered expression area, and could either be cell intrinsic, or too small to visualize by means of in situ hybridization. However, the expression area of *Tph2* (Figure 7E,E’) and *Wnt8b* (Figure 7F) appeared to be altered in the *Nkx2.9*LacZ/LacZ animals at E14.5. The strong separation of *Tph2* and *Th* as seen in the *Nkx2.9*+/+ was no longer clearly visible in the *Nkx2.9*LacZ/LacZ at E14.5 (Figure 7E’ pseudo-overlay *Th* (green) and *Tph2* (red) white arrowheads). The expression of *Tph2* appeared to be entering the isthmic area and directly borders, or even overlapped with, the most caudal part of *Th* expression. Furthermore, the expression area of *Wnt8b* appeared to be decreased in the midbrain of *Nkx2.9*LacZ/LacZ embryos compared to *Nkx2.9*+/+ at E14.5, consistent with the decrease in expression of *Wnt8b* as seen in the RNA-sequencing.

Based on the transcriptome analysis and expression data of *Tph2* and *Wnt8b*, *Nkx2.9* could have a function in establishment of the IsO and repression of hindbrain-specific cell-fate. As the IsO is already long established after E14.5, we set out to determine whether early patterning of the IsO is affected upon loss of *Nkx2.9*. To this end, we examined the expression of *Wnt1*, an important regulator of the IsO and midbrain area [32,33,34], and the expression of *Fgf8*, the quintessential marker of the IsO [35,36] in the *Nkx2.9*-ablated midbrain at E12.5. We observed no apparent changes in the expression of either *Wnt1* (Figure 7H) or *Fgf8* (Figure 7I) in the absence of *Nkx2.9*. Taken together, these data suggest that *Nkx2.9* does not function in the early establishment of the IsO, but rather has a later role in repression of hindbrain fate in the isthmic area.

## 3. Discussion

We showed that *Nkx2.9* is lost in *En1* mutant embryos and that expression of *Nkx2.9* overlaps with the expression of known IsO markers, such as *Fgf8*. Furthermore, we established that a majority of mdDA neurons express *Nkx2.9* at some point during development. Although *Nkx2.9* is expressed in a large subpopulation of mdDA neurons, and loss of this gene leads to a temporary loss of *Dat* in mdDA neurons, it does not affect the amount or identity of mature mdDA neurons. However, *Nkx2.9* may have a role in repression of a hindbrain fate and activation of a midbrain fate, as the expression of *Thp2*, a gene involved in establishment of the serotonin system, is upregulated in the isthmic area, and expression of *Wnt8b* is downregulated in the VZ of the midbrain floor plate in these mutants. Taken together, the results suggest a minor role for *Nkx2.9* in midbrain development, by suppression of a hindbrain cell fate and an initial decrease in the expression of specific mdDA (subset) markers in the midbrain, which is normalized in the adult. This work could benefit research on dopamine progenitor formation and maturation of pluripotent stem cells in vitro, as it sheds more light on the underlying molecular cascade of mdDA neuronal maturation.

This study provides additional information on the intricate network of transcriptional regulation that is required for a progenitor neuron to develop into a bona fide mdDA neuron. Interestingly, although *Nkx2.9* showed restricted expression in the isthmic area of the mid- and hindbrain, many mdDA neurons are derived from this lineage, as shown by the expression of LacZ. However, the fact that not all mdDA neurons showed LacZ expression suggests that within the mdDA neuronal population there are neurons belonging to a subset that expressed *Nkx2.9* and neurons that did not. Recently, more attention has been given to the identification of different molecular subsets within the mdDA neuronal pool in both the murine and human brain. Studies performed by Poulin et al. (2014) and La Manno et al. (2016) made use of single-cell RNA-sequencing to obtain more insight into the different molecular subsets within the mdDA neuronal population [37,38]. It would be interesting to determine to which subset these neurons belong, or whether they are equally divided over the different subsets. 

Prior research from our group revealed that *Dat* expression is driven by the activity of *Nurr1*, *Pitx3*, and *En1* [16,39]. In the current work, we showed that *Nkx2.9* is also involved in correct programming, leading to the correct expression and timing of *Dat* expression in mdDA neurons. It is possible that the delay in *Dat* expression is a result of a cell-intrinsic alteration of *En1* expression, as this marker was found to be upregulated in the transcriptome analysis of *Nkx2.9*-ablated animals. Expression of *Nurr1* (Appendix A) and *Pitx3* (Appendix A) was found to be unaltered in *Nkx2.9* mutants, suggesting that the role of *Nkx2.9* on *Dat* expression may not be caused by altered expression of either *Nurr1* or *Pitx3*. *Dat* expression is also negatively regulated by *Otx2* [40]; however, the area of expression or *Otx2* transcript levels are not altered (Appendix A), suggesting that the *Nkx2.9* pathway to *Dat* expression is also independent of *Otx2*. In addition to the evident change in *Dat* expression, we observed a slight deficiency in *Cck* expression in *Nkx2.9*-ablated animals. *Cck* is a well-known marker of caudal (primordial VTA) mdDA neurons [16]. However, although the changes in expression of *Dat* and *Cck* were apparent in in situ experiments, these changes were not detected in the RNA-sequencing on E14.5 embryonic midbrains. This could be because the RNA-sequencing does not have enough sequencing-depth to actually pick up on these changes, or there is a timing difference between the analyzed embryos. If expression of both *Dat* and *Cck* has just started at E14.5, small differences in timing could result in a large inter-embryonic variation. Taken together, these data further suggest that initial subset programming might be influenced by *Nkx2.9*. However, as expression of *Wnt8b* is affected in the *Nkx2.9* mutant, we cannot rule out the possibility that the effects on *Dat* and *Cck* expression at E14.5 may be caused by affected WNT-signaling in this area, considering that a subset of *Dat*-expressing cells is responsive for canonical Wnt-signaling [32]. Interestingly, in zebrafish, it has been shown that *Wnt8b* is involved in establishment of the IsO, and loss of *Wnt8b* results in an aberrant number of dopamine neurons in the zebrafish diencephalon [25,27]. The fact that loss of *Nkx2.9* leads to a downregulation of *Wnt8b* and loss of repression of a hindbrain-specific cell fate in the isthmic area, could point to a role in IsO maintenance by regulation of Wnt8b expression in the murine midbrain. 

The transient nature of the change in *Dat* expression, and no obvious effect on TH or other mdDA neuronal markers, may suggest that *Nkx2.9* interacts in a (partly) redundant molecular network, in which the loss of *Nkx2.9* can be compensated, albeit with a possible delay. One factor that could be involved in this compensation is *Nkx2.2*. As described previously, the expression patterns of *Nkx2.9* and *Nkx2.2* are highly similar, and function together in floor plate development of the spinal cord [8,9]. The fact that both *Nkx2.9* and *Nkx2.2* function in development of the serotonin system in the hindbrain could point to a similar function of both genes [7,31]. Similar transcriptional redundancies have been previously described for transcription factors that code early dopaminergic precursors, such as *Mash1* and *Ngn2* [41]. Recently it was shown that during lung cancer development *Nkx2.9* interacts with *Nkx2.1* and *Pax9* [42], and while these two factors are not expressed in mdDA (precursor) neurons, it is possible that *Nkx2.9* maintains a similar molecular relationship in mdDA neurons with other (yet unknown) transcription factors. 

Taken together, the current work clearly establishes that many but not all mdDA neurons originate from an *Nkx2.9*-positive precursor lineage. Moreover, *Nkx2.9* appears to contribute to the network of transcription factors that promote terminal differentiation of mdDA neurons and is involved in repressing a hindbrain-fate in the isthmic area. 

## 4. Materials and Methods

### 4.1. Animals

*En1*tm1Alj/+ animals were back-crossed to the C57BL6/J line generating *En1*+/+ (WT), *En1*tm1Alj/+ (Het) and viable *En1*tm1Alj/tm1Alj (KO) offspring, and genotyped using specific primers [16,19,43]. *Nkx2.9*LacZ/LacZ and *Nkx2.9*LacZ/+ animals were back crossed in C57BL/6 mice and genotyped using specific primers, as described previously [22].

Embryos were isolated at embryonic day (E)12.5 and E14.5 considering the morning of detection of the vaginal plug as E0.5. Adult brains were isolated from animals at 2.5 or 6 months old (including age-matched controls). 

The study was conducted according to the guidelines of the Declaration of Helsinki and approved by the Dutch Ethical Committee of the UvA (AVD111002016742, 20 January 2017) and the University of Pittsburgh Animal Care and Use Committees.

All experiments were performed on two to three embryos/brains from one to two different litters. Quantification experiments were performed on at least three brains per genotype from at least two litters (numbers are stated per quantification).

### 4.2. qPCR

Relative expression levels were determined by qPCR real-time PCR (Lightcycler 480, Roche, Bazel, Switzerland) using the QuantiTect SYBR Green PCR LightCycler Kit (QIAGEN, Hilden, Germany) according to the manufacturer’s instructions. For each reaction, 10 ng (dissected midbrain) of total RNA was used as input. Primer sequences: 18S FP 5′-AAACGGCTACCACATCCAAG-3′ RP 5′-CCTCCAATGGATCCTCGTTA-3′, and *Otx2* FP 5′-CCAGGGTGCAGGTATGGTTTA-3′ RP 5′-CTTCTTGGCAGGCCTCACTT-3′.

### 4.3. RNA-seq

RNA-seq libraries were produced as described previously [44]. Briefly, RNA was extracted from dissected E14.5 midbrain material from *Nkx2.9*+/+, *Nkx2.9*LacZ/+ and *Nkx2.9*LacZ/LacZ animals using the miRNeasy Mini Kit (QIAGEN, Hilden, Germany), followed by ribosomal RNA depletion using the RiboMinus Eukaryotic System V2 (Life Technology, Thermo Scientific, Waltham, MA, USA). Upon synthesis of cDNA for each individual midbrain, two midbrains per genotype were pooled to create one indexed library using a TruSeq Nano DNA Sample Preparation kit (Illumina, San Diego, CA, USA). In total nine libraries were produced (*n* = 3 per genotype). 100-bp paired-end sequencing was carried out using an Illumina HiSeq 2500 instrument (San Diego, CA, USA), and the sequence reads were aligned (Hisat2) against the mouse genome GRCm38/mm10 [45]. Reads were counted and normalized (featurecount, Deseq2) and ranked by statistical value (Deseq2). The RNA sequencing dataset generated and analyzed during the current study are available in the GEO repository (GSE185584).

### 4.4. In Situ Hybridization (ISH)

In situ hybridization was performed as described previously [46]. Digoxigenin-labeled probes for *Th*, *Fgf8*, *Dat*, *Pitx3*, *Nurr1*, *Wnt1*, *Ahd2*, *Cck*, *En1*, *En2*, and *Otx2* were used as previously described [16,19,47,48,49,50,51]. Additional probes were as follows: *Nkx2.9* was designed at bp 642–1070 (428 bp length) of the mouse cDNA sequence (NM_008701.2), *Pax7* was designed at 866–1263 (398 bp length) of the mouse cDNA sequence (NM_011039.2), *Tph2* was designed at 1021–1460 (440 bp length) of the mouse cDNA sequence (NM_173391.3), and *Wnt8b* was designed at 35–470 (436 bp length) of the mouse cDNA sequence (NM_011720.3).

### 4.5. Histology and Fluorescent Immunohistochemistry

Embryos were fixed in 4% paraformaldehyde (PFA) in PBS, cryoprotected in 30% sucrose in PBS and subsequently stored at −80 °C. Sagittal sections (16 µm) were cut on a cryostat, after which they were washed with TBS and blocked in 4% Fetal Calf Serum (FCS) in THZT (50 mM Tris-HCl pH 7.6, 0.5 M NaCl, 0.5% Triton). After another wash treatment with TBS, sections were incubated overnight at 4 °C with primary antibody in THZT. Sections were washed three times (TBS) the following morning and incubated for a minimum of 2 h at room temperature (RT) with secondary antibody in TBS, followed by a wash treatment with PBS. DAPI staining was performed (1 mg/mL 1:5000) for 5 min, after which sections were washed with PBS. Finally, sections were embedded with Fluorsave (Calbiochem, Amsterdam, The Netherlands). Primary antibodies used were Rabbit α-TH (Pel-freez, Rogers, AR, USA 1:1000), Rabbit α- β-galactosidase (Thermo Scientific, Waltham, MA, USA MP 559761, 1:1000), Rabbit α-phosphorylated Histone H3 (Abcam, Cambridge, UK, 1:1000). Secondary antibodies that were used were Goat α-Rabbit Alexa 555 (1:1000), Goat α-Rabbit Alexa 488 (1:1000), Goat α-Chicken Alexa 488 (1:1000), all from Invitrogen (Waltham, MA, USA).

Adult brains (6 months) were fixed in 10% buffered neutral formalin, dehydrated, and embedded in paraffin. Coronal sections (5 µm) were cut on a microtome. Sections were deparaffinized and rehydrated (xylene and graded ethanol series from 100% to 50% and finally saline) and were subjected to an antigen retrieval step as follows. Slides were incubated with 0.1 M citrate buffer pH6 for 3 min at 800 W and 9 min at 400 W, cooled down to RT in a water bath.

### 4.6. DAB Immunohistochemistry

Sections (16 μm) were post-fixed in 4% PFA for 30 min then DAB (3,3Ј-diamino-benzidine) staining was performed as previously described [52]. Antibodies were used as followed: Rabbit α-TH (Pel-freez, Rogers, AR, USA 1:1000), Goat α-Rabbit-biotinylated (Vector Laboratories, Burlingame, CA, USA 1:1000).

### 4.7. X-galactosidase Staining Protocol

Fresh frozen sections, whole brains or hemitubes were post-fixed with 4% PFA for 45 min. The tissue was washed with PBS three times, and once with staining solution (5 mM potassium ferricyanide, 5 mM potassium ferrocyanide, 2 mM MgCl_2_ in PBS). Tissue was incubated at 4 °C with staining solution complemented with 1 mg/mL X-galactosidase (X-gal) overnight, protected from light, until staining was optimal. Sections were then rinsed in PBS, dehydrated and embedded in Entellan (Merck, Darmstadt, Germany). Whole brains and hemitubes were rinsed in PBS, and cleared in 25%, 50% and 80% glycerol in PBS (Applied Biosystems, Waltham, MA, USA).

## Figures and Tables

**Figure 1 ijms-22-12663-f001:**
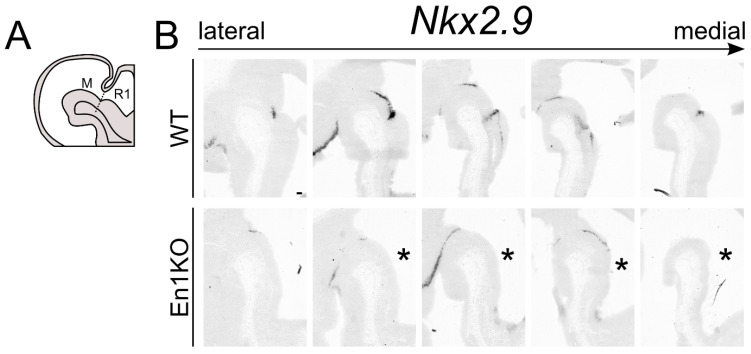
*Nkx2.9* is absent in the IsO of *En1*-ablated animals. (**A**) Schematic, sagittal section of the mouse midbrain (M) and rhombomere 1 (R1) at developmental stage E12.5. The position of the IsO is indicated by a dotted line. (**B**) In situ hybridization for *Nkx2.9* on different sections from lateral to medial in the developing midbrain region in wild type (WT) and *En1*-ablated (*En1*KO) animals at E12.5. Scale bar represents 100 µm. Asterisk indicates missing *Nkx2.9* expression.

**Figure 2 ijms-22-12663-f002:**
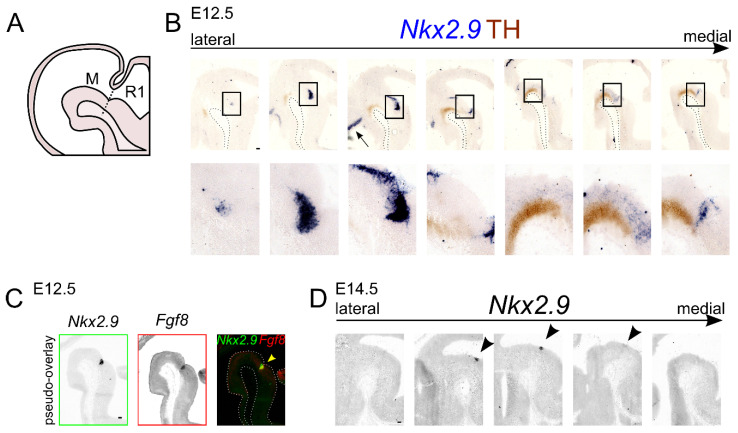
*Nkx2.9* is expressed in the mdDA progenitor region at E12.5. (**A**) Schematic, sagittal section of the mouse midbrain (M) and rhombomere 1 (R1) at developmental stage E12.5. The position of the IsO is indicated by a dotted line. (**B**) In situ hybridization experiment reveals that *Nkx2.9* expression (blue) is present in the diencephalon, mesencephalon and metencephalon. In medial sections, the *Nkx2.9* transcript is found at the IsO, and it extends dorsolaterally towards the diencephalon into the ZLI (arrow). Black boxes indicate areas of magnification. TH immunohistochemistry (DAB staining, brown) represents the location of mdDA neurons. (**C**) Expression of *Nkx2.9* (green box) overlaps with *Fgf8* transcript (red box) in medial midbrain at E12.5, as is indicated in a pseudo-overlay (arrowhead). (**D**) *Nkx2.9* in situ hybridization experiment on sagittal brain sections from stage E14.5 shows that it is limited to a small area in the caudomedial midbrain, close to the IsO (arrowheads). Scale bar represents 100 µm.

**Figure 3 ijms-22-12663-f003:**
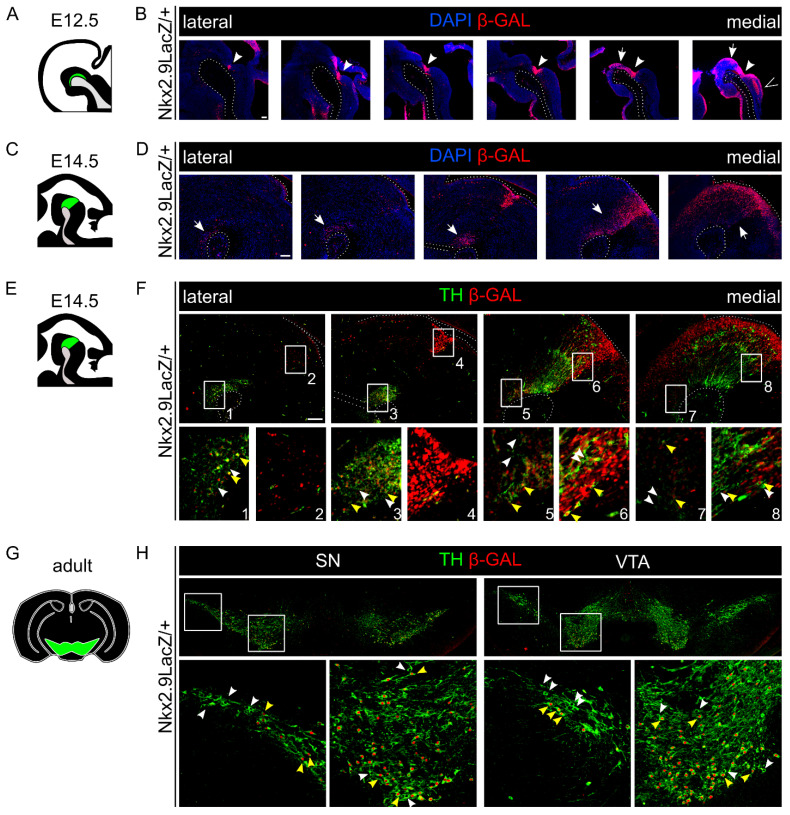
*Nkx2.9*LacZ-fate-mapping reveals that *Nkx2.9* expression was encountered by a large portion of mdDA neurons. (**A**) Schematic, sagittal section of the mouse midbrain at E12.5; the mdDA area is indicated in green. (**B**) Immunohistochemistry for β-gal (red) and nuclei (DAPI, blue) on sagittal E12.5 brain sections shows that the β-gal protein is present in the IsO (arrowhead), the mdDA area of the midbrain (arrow) and in the hindbrain (>). (**C**) Schematic, sagittal section of the mouse midbrain at E14.5; the mdDA area is indicated in green. (**D**) Immunohistochemistry for β-gal (red) and nuclei (DAPI, blue) on sagittal E14.5 brain sections shows that the β-gal protein is present in the IsO region (arrowhead), the mdDA area of the midbrain (arrow) and in the hindbrain (>). (**E**) Schematic, sagittal section of the mouse midbrain at E14.5; the mdDA area is indicated in green. (**F**) Immunohistochemistry for TH (green) and β-gal (red) at E14.5 in sagittal brain sections. Double-positive TH+/β-gal+ neurons (yellow arrowheads) and single TH+/β-gal neurons (white arrowheads) are present in the lateral and medial mdDA region (**G**) Schematic, coronal section of the adult mouse brain; the mdDA area is indicated in green. (**H**) Immunohistochemistry for TH (green) and β-gal (red) in adult brain sagittal sections show that double-positive TH+/β-gal+ neurons (yellow arrowheads) and single TH+/β-gal-neurons (white arrowheads) are present in SNc and VTA in adult brains. White dotted line in B, D and F marks the borders of the tissue. White boxes indicate areas of magnification in each panel. Scale bar represents 100 µm.

**Figure 4 ijms-22-12663-f004:**
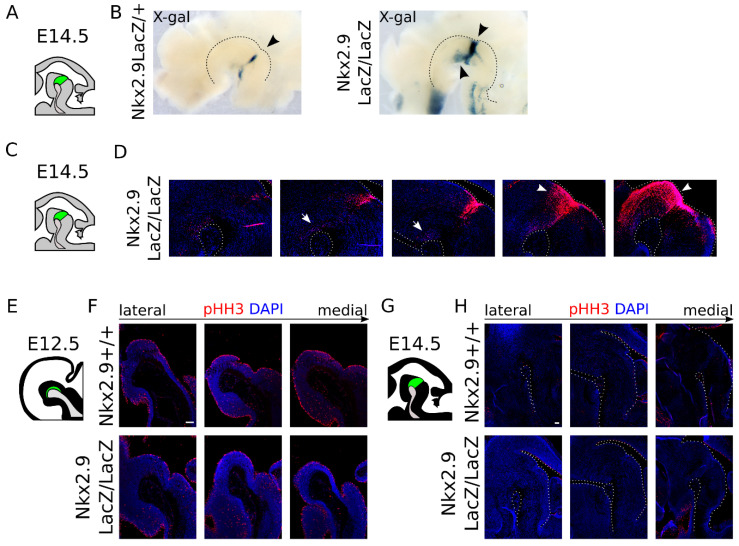
*Nkx2.9* is strongly expressed in the IsO at E14.5 but does not affect proliferation in this area. (**A**) Schematic, sagittal section of the mouse midbrain at E14.5; the mdDA area is indicated in green. (**B**) Medial view of dissected hemitube at E14.5. β-gal activity (visualized by X-gal staining) is present in the IsO-area in control hemitube (arrowhead). β-gal activity is markedly increased in *Nkx2.9*-ablated animals in the IsO area and in midbrain (arrowheads). Black dotted line delineates the mesencephalic flexure. (**C**) Schematic, sagittal section of the mouse midbrain at E14.5; the mdDA area is indicated in green. (**D**) Staining of β-gal protein (red) and nuclei (DAPI, blue) shows increased levels of β-gal protein are present in the (sub)ventricular zone in the midbrain at E14.5 in *Nkx2.9*LacZ/LacZ animals (arrowheads). Note that β-gal staining demonstrated decreased β-gal staining in the mdDA neuronal region in *Nkx2.9*LacZ/LacZ animals (arrows). White dotted line marks the borders of the tissue. (**E**) Schematic, sagittal section of the mouse midbrain at E12.5; the mdDA area is indicated in green. (**F**) Examination of medial to lateral sections of immunohistochemistry experiment for phosphorylated Histone H3 (pHH3, red) and a nuclear staining (DAPI, blue) demonstrates no clear alteration in proliferation in the midbrain region at E12.5 in *Nkx2.9*LaczZ/LacZ animals compared to litter mate *Nkx2.9*LacZ/+ controls. (**G**) Schematic, sagittal section of the mouse midbrain at E14.5; the mdDA area is indicated in green. (**H**) Examination of medial to lateral sections of immunohistochemistry experiment for phosphorylated Histone H3 (pHH3, red) and a nuclear staining (DAPI, blue) demonstrates no obvious alteration in proliferation in the midbrain region at E14.5 in *Nkx2.9*LaczZ/LacZ animals compared to litter mate *Nkx2.9*LacZ/+ controls. Scale bar represents 100 µm.

**Figure 5 ijms-22-12663-f005:**
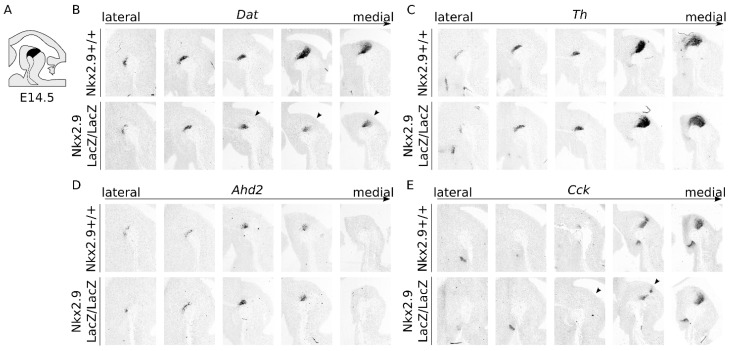
*Dat* and *Cck* expression is down-regulated in a caudomedial area of mdDA neurons in *Nkx2.9*-ablated animals at E14.5 (**A**) Schematic, sagittal section of the mouse midbrain at E14.5; the mdDA area is indicated in black. (**B**) Lateral to medial sections of *Dat* in situ hybridization experiments reveals that *Dat* expression is down-regulated in caudomedial mdDA neurons (arrowheads) in *Nkx2.9*-ablated midbrains at E14.5. (**C**) *Th* expression analysis on adjacent sections reveals no aberrations in *Nkx2.9*-ablated midbrains at E14.5. (**D**) In situ hybridization for the rostral marker *Ahd2* indicates that the expression pattern is unaffected as a consequence of *Nkx2.9*-ablation. (**E**) Expression analysis for the caudal marker *Cck* on adjacent sections indicates that *Cck* expression is slightly affected in the caudo-medial area of *Nkx2.9*-ablated midbrains (arrowheads).

**Figure 6 ijms-22-12663-f006:**
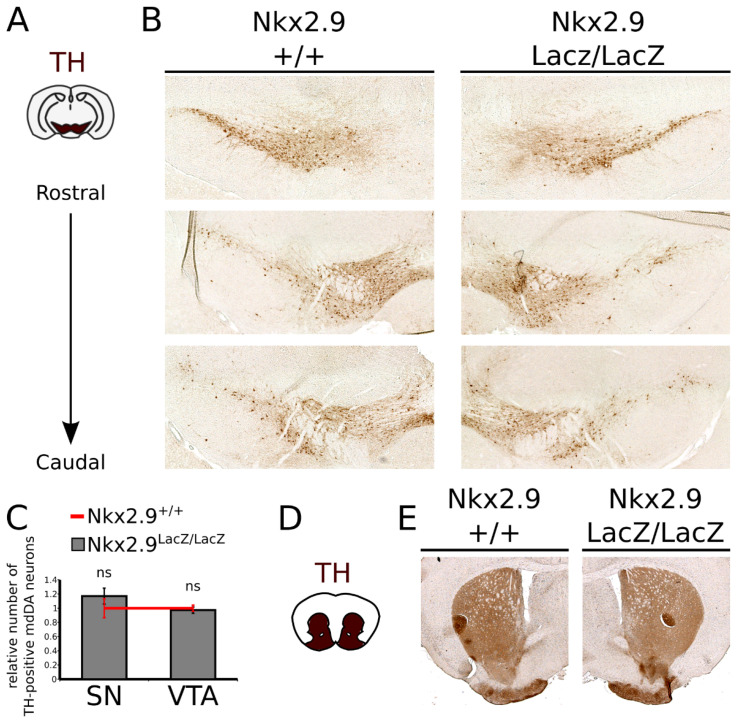
*Nkx2.9*-ablation does not affect the cytoarchitecture and number of mdDA neurons in the adult midbrain. (**A**) Schematic, coronal section of the adult mouse brain; the mdDA area is indicated in brown. (**B**) TH immunohistochemistry on adult coronal sections of the midbrain, at several views from rostral to caudal. The cytoarchitecture of the TH-expressing mdDA neurons is not altered in the adult midbrain of *Nkx2.9*-ablated mice. (**C**) Quantification of TH-expressing mdDA neurons in *Nkx2.9-*ablated adult brains and age-matched controls revealed no significant (ns) changes in the number mdDA neurons (*n* = 3). (**D**) Schematic, coronal section of the adult mouse brain; the striatum is indicated in brown. (**E**) Coronal section of the striatum stained for TH protein (brown), reveals that the TH-density is similar in WT and *Nkx2.9* mutant animals.

**Figure 7 ijms-22-12663-f007:**
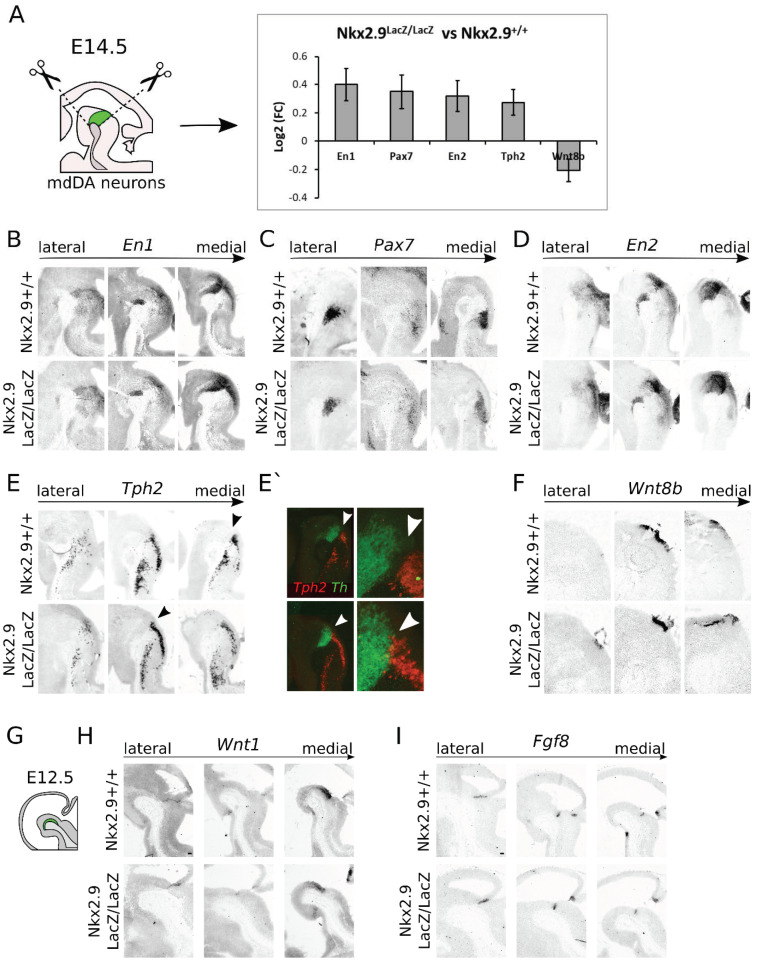
Next-generation RNA sequencing of *Nkx2.9* mutant midbrain material compared to wild-type controls (**A**) Schematic representation of the RNA-sequencing setup. RNA from dissected midbrains at E14.5 of *Nkx2.9*-ablated animals and controls were isolated to produce RNA-seq libraries. Relative expression levels of differentially regulated transcripts of interest, related to mid- or hindbrain development, *En1*, *Pax7*, *En2*, *Tph2*, and *Wnt8b*. (**B**) Expression area of *En1* is not altered in the midbrain of E14.5 *Nkx2.9*LacZ/LacZ embryos, compared to WT littermates. (**C**) Expression area of *Pax7* is not altered in the midbrain of E14.5 *Nkx2.9*LacZ/LacZ embryos, compared to WT littermates. (**D**) Expression area of *En2* is not altered in the midbrain of E14.5 *Nkx2.9*LacZ/LacZ embryos, compared to WT littermates. (**E**) The expression area of *Tph2* extends more rostrally to the caudal part of the midbrain of E14.5 *Nkx2.9*LacZ/LacZ embryos, compared to WT littermates. (black arrowheads) (**E’**) Pseudo-overlay of *Th* (green) and *Thp2* (red) expression clearly shows that the strong separation of the two expression areas is lost in the isthmic area of the *Nkx2.9* ablated animals, resulting in a more rostral expression of *Thp2* bordering, and even intruding, the caudal Th expression area (white arrowheads). (**F**) Expression area of *Wnt8b* appears to be slightly altered in the midbrain of E14.5 *Nkx2.9*LacZ/LacZ embryos, compared to WT littermates, suggesting a possible effect on regulation of WNT-signaling of *Nkx2.9*. (**G**) Schematic, sagittal section of the mouse midbrain at E12.5; the mdDA area is indicated in green. (**H**) *Wnt1* transcript expression revealed no altered expression patterns in *Nkx2.9*-ablated embryos at E12.5. (**I**) *Fgf8* transcript expression revealed no altered expression patterns in *Nkx2.9*-ablated embryos at E12.5.

## Data Availability

The data presented in this study are openly available in the GEO repository (GSE185584).

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
