# Peer review of "Nkx2.9 Contributes to Mid-Hindbrain Patterning by Regulation of mdDA Neuronal Cell-Fate and Repression of a Hindbrain-Specific Cell-Fate"

_ijms, 2021, doi:10.3390/ijms222312663_

Round 1

Reviewer 1 Report

In this study, the authors present a thorough investigation of the role of Nkx2.9 on mid- and hindbrain development in the rodent. After an inclusive series of staining experiments and transcriptomics using both wild-type and Nkx2.9-deleted lines, the authors conclude that Nkx2.9 does have an impact on this aspect of subcortical development, gaining some insight into the maturation of the mesodiencephalic dopaminergic neuron population. However, the phenotype was extremely mild, and the nature of the mechanism by which Nkx2.9 contributes to the progression of this neuronal population is still largely unanswered. Despite these limitations, the authors were thorough enough in their studies that the findings, though slight, come across as highly convincing.

Minor criticisms:

  • No indication as to the number of animals used for the various ISH/IHC experiments
  • The Nkx2.9 and Fgf8 images in Figure 2C are out of register, making it hard to compare the location of each stain in relation to each other.
  • Slight grammatical issues and typos (e.g. “ventral segmental area” [Line 45], “noteworthy” used as an adverb [Line 151])

Reviewer 2 Report

In the manuscript Nkx2.9 contributes to mid- hindbrain patterning by regulation of the mdDA neuronal cell-fate and repression of a hindbrainspecific cell-fate” the authors aim to determine the role of Nkx2.9  during  mdDA neuron development. Authors detect Nkx2.9 in the IsO as well as in the VZ and SVZ and show that Nkx2.9 expression is lost in En1 mutant embryos. By combining loss of function experiments with a comprehensive  immunohistochemical analyses, the authors clarify the role of Nkx2.9 during mid-hindbrain patterning and mdDA neuronal subtype specification.

The manuscript presents novel elements with detailed characterization of Nkx2.9 in mouse midbrain development. The manuscript is overall well organized, and conclusions are well supported by the results.

Minor comments:

  • The authors claim that "a major subset of mdDA neurons originates from a Nkx2.9-positive lineage". To better substantiate this statement, the authors should include immunohistochemistry analysis for FOXA2 or LMX1a/b in Nkx2.9 knock-out/LacZ knock-in mouse model at E12.5.

  • In order to draw firm conclusions that loss of Nkx2.9 does not impact on the mdDA neuronal subtype specification, authors should provide Girk2 and Calb expression analyses in Nkx2.9 +/+ and 9 LacZ/LacZ at E14.5.

  • Single cell sequencing has now provided novel insights into developing and maturation of mouse midbrain . The authors may briefly comment the usefulness of single cell sequencing to better uncover molecular mechanism underlying dopamine neuronal differentiation in the light of recent studies (La Manno et al., Cell, 2017, Tiklova et al., Nat Comm, 2019).

  • Authors should also briefly discuss the current use of pluripotent stem cell as a powerful tool to reconstruct dopamine progenitor formation and maturation in a dish.
